# Optimization of Scanning Protocol for AI-Integrated Assessment of HER2 Dual Bright-Field In-Situ Hybridization Application in Breast Cancer

**DOI:** 10.3390/bioengineering12060569

**Published:** 2025-05-26

**Authors:** Nilay Bakoglu Malinowski, Takashi Ohnishi, Emine Cesmecioglu, Dara S. Ross, Tetsuya Tsukamoto, Yukako Yagi

**Affiliations:** 1Department of Pathology and Laboratory Medicine, Memorial Sloan Kettering Cancer Center, New York, NY 10065, USA or nilay.bakoglu@medipol.com.tr (N.B.M.); t.ohnishi0417@gmail.com (T.O.); rossd@mskcc.org (D.S.R.); tsukamt1@mskcc.org (T.T.); 2Department of Pathology, Faculty of Medicine, Istanbul Medipol University, 34214 Istanbul, Turkey; 3Department of Pathology, Marmara University Research and Education Hospital, 34899 Istanbul, Turkey; eminecesmecioglu@gmail.com

**Keywords:** artificial intelligence, dual bright-field in-situ hybridization, breast carcinoma, whole slide image, scanning protocol

## Abstract

Accurately determining HER2 status is essential for breast cancer treatment. We developed an AI-integrated in-house application for automated Dual bright-field (BF) in situ hybridization (ISH) analysis on whole slide images (WSIs), although optimal scanning conditions remain unclear. We evaluated scanners and optimized scanning protocols for clinical application. Ten de-identified invasive breast carcinoma cases, with HER2 immunohistochemistry and FISH results, were analyzed using three scanners and six scanning protocols. WSIs scanned by Scanner ‘A’ have 0.12 µm/pixel with 0.95 NA (A1) and 1.2 NA (A2); Scanner ‘B’ have 0.08 µm/pixel (B1); 0.17 µm/pixel (B2); and 0.17 µm/pixel with extended focus (1.4 µm step size and three layers) (B3); Scanner ‘C’ has 0.26 µm/pixel (C1) resolution. Results showed scanning protocols A1, A2, B2, and B3 yielded HER2 gene amplification status and ASCO/CAP ISH group results consistent with manual FISH as the ground truth. However, protocol C demonstrated poor concordance due to nuclei detection failure in six cases. The AI-integrated application achieved the best performance using scanning protocols with optimized resolutions of 0.12 µm/pixel and 0.17 µm/pixel with extended focus. This study highlights the importance of scanner selection in AI-based HER2 assessment and demonstrates that optimized scanning parameters enhance the accuracy and reliability of automated Dual BF ISH analysis.

## 1. Introduction

Human epidermal growth factor receptor 2 (HER2) is a receptor tyrosine kinase, a member of the epidermal growth factor receptor family, and encoded by the *ERBB2* gene located on chromosome 17 at 17q12 [1]. HER2 protein overexpression and/or gene amplification are seen in 15–20% of breast carcinomas [2]. Determination of HER2 status has an important role in the treatment process and prognosis of breast cancers [3]. In situ hybridization (ISH) and immunohistochemistry (IHC) are the most common methods for HER2 testing [4,5,6]. Fluorescence in situ hybridization (FISH) is considered the gold standard because it is a very sensitive assay for detecting HER2 amplification and it is accepted as a reference test because it predicts the response to trastuzumab more accurately. However, it has led to the search for alternative tests because it requires a modern and expensive fluorescence microscope and multi-band-pass fluorescence filters, and the fluorescence fades so quickly that it cannot provide a permanent record. Alternative ISH methods can overcome some of these disadvantages by using conventional bright field (BF) microscopy, which provides permanent records and allows for the visualization of histology [7]. Dual BF ISH is based on quantifying the number of Chromosome Enumeration Probe 17 (CEP17) and HER2-specific signals under the BF microscope [8] and it is reproducible. However, pathologists need to count the numbers of CEP17 and HER2-specific signals in 20 or more cells, which is a time-consuming task [9].

The growth of whole slide imaging (WSI) technology and artificial intelligence (AI) based image analysis applications aspire to achieve an automatic pathological diagnosis and smoother workflow [10]. Over the last decade, a wide variety of WSI scanners have been developed and commercialized. These commercial WSI scanners have differences in imaging node, slide capacity, scan speed, image magnification and resolution, digital slide format, multilayer capability, and barcode support. The image analysis on WSI offers practical solutions in quantitative and objective studies [3,4,11]. Image analysis algorithms will help pathologists determine ideal areas of interest for Dual FISH and/or Dual BF ISH tests, then segment nuclei and count each signal automatically. Several AI-integrated analysis applications have been applied for ISH quantification [12,13,14].

Despite increased interest in the development of automated FISH signal quantitative assessment, currently, there are no commonly used protocols or standardized quantitative assessment methods [15,16,17].

In our previous studies [12,18], our in-house application by one scanner and one scan profile showed 94% (33/35) concordance with manual American Society of Clinical Oncology (ASCO)/College of American Pathologists (CAP) ISH group results. From the results, we found that the reasons for discordance stemmed from unclear nucleus border/signal, overlapping nuclei, and focusing issues. Other WSI scanners or scanning protocols might improve the sharpness of the nucleus border and signal and the robustness of the image focus.

In this study, we compared the image quality scanned by different WSI scanners and scanning protocols and the analytical results with the in-house application to provide the best analytical results by image analysis algorithms.

## 2. Materials and Methods

### 2.1. Cases and Slide Preparation

Ten formalin-fixed paraffin-embedded (FFPE) block samples from 10 de-identified invasive breast carcinoma cases were selected from our database (Table 1). Five or six serial sections of 4 µm thickness were prepared from each sample, and then the hematoxylin and eosin (H&E) as well as HER2 IHC, Dual FISH, and Dual BF ISH-stained slides were prepared. Two-color Dual FISH analysis was performed using the HER2 IQ Dual FISH pharmDx assay probes and procedure (Dako, Glostrup, Denmark).

### 2.2. Dual BF ISH

Dual BF ISH testing was performed using VENTANA HER2 Dual ISH DNA Probe Cocktail assay (Roche Tissue Diagnostics, Tucson, AZ, USA) as described by the manufacturer. Briefly, the slides were deparaffinized and followed by dual-hybridization steps; the dinitrophenyl-linked HER2 probe was visualized by an insoluble precipitate of silver chromogen, and the visualization of the digoxigenin-linked Chr17 probe was detected by the soluble precipitate of the alkaline phosphatase-based Fast Red chromogenic system. The slides were counterstained with hematoxylin for the visualization of the nuclear morphology, rinsed in distilled water, dried, and cover-slipped.

### 2.3. Regions of Interest (ROI)

Six different protocols were used for 10 cases and a total of 60 WSIs were created, from which 461 ROIs were selected, uploaded, analyzed, and evaluated. Fifty of the 60 WSIs were scanned by scanner A and B protocols and 243 specific ROIs, averaging 3–4 ROIs per WSI, were selected by pathologists and analyzed by the AI application. The remaining 10 of the 60 WSIs were scanned by scanner C protocol, and 218 specific ROIs, averaging 10–20 ROIs per WSI, were selected and analyzed.

All scanned images were viewed using a Slide Viewer (3DHISTECH. LTD, Budapest, Hungary), which supports all image formats we used. First, invasive tumor and in situ areas were annotated on H&E slides by pathologists, and the annotations were cloned to the WSIs of Dual BF ISH. Second, specific ROIs for analysis were drawn on WSIs of Dual BF ISH. The ROIs were copied to the corresponding position in the WSIs of all scanning protocols if the image quality was equivalent for Scanners A and B (Figure 1). For the WSIs by Scanner C, ROIs were drawn manually by pathologists separately. Selection of ROI was started with 1 or 2 ROIs equivalent to 0.1 to 0.3 mm^2^ area. When the required number of nuclei for reporting could not reach 20 in the ROIs, additional ROIs were added. Once the annotation was made, the WSIs were loaded into the AI-integrated Dual BF ISH HER2 counting application. For borderline or heterogenous cases, the pathologist selected a second non-overlapping ROI to count another 20 nuclei if there were not enough nuclei to reach 40 in the initial ROI.

### 2.4. WSI Scanner and Scanning Protocol

Three WSI scanners A, B, and C were used for slide digitization, and a total of six scanning protocols A1 and A2, B1–B3 and C1 were tested for this study as detailed in Table 2. For the scanning protocols A1 and A2, a dry lens or a water immersion lens were selected, respectively. For B1–3, Scanner B was operated with dry objective lenses of 40× (B1), 20× (B2), and 20× with extended focus (B3). Scanner C had a fixed 40× dry lens. Cases with unacceptable artifacts and out-of-focuses were subjected to re-scanning for sufficient image quality.

Scanner A is an ultra-high throughput full-slide imaging scanner designed to meet the demands of large-scale laboratories and research institutions. It provides multi-layer scanning for better depth and clearer analysis of complex cytological samples. It provides ultra-sharp imaging with enhanced resolution and contrast, ideal for detailed applications such as cytology and immunohistochemistry. Scanner B is a high-speed digital slide scanner that offers a 300-slide capacity and supports both bright-field and fluorescence scanning. It features AI-powered texture detection and next-generation automation such as Z-stack scanning and autofocus scanning. The lens on Scanner C features extra-wide flat field correction, providing a much larger field of view (1 mm) that can accommodate extremely large digital images for fast scans and a resolution of 0.26 μm/pixel at 40× magnification. Real-time focusing enables digital image capture that combines an imaging line sensor and a focusing line sensor.

### 2.5. AI-Integrated Dual BF ISH Application

In our study, we used the deep learning AI-based application that was proposed as an in-house application (Memorial Sloan Kettering, New York, NY, USA) in our department in the previous study and that enables the extraction and selection of non-handcrafted features by a convolutional neural network (CNN). This application applies a non-linear support vector machine (SVM) binary classifier to select individual cores using nine shape and boundary-based features. For nuclear detection, 10-fold cross-validation was performed using 3020 cores from 15 different cancer regions in 7 WSIs, each from a different cancer case [19]. For this study, CNN is used for the segmentation of the nucleus and each signal. The basic network architecture is U-Net which is one of the CNN-based architectures developed for image segmentation. Convolution layers and activation layers on the typical U-Net were modified by the engineer. Ten WSIs with annotation were used. From each of them, 176 images with 512 × 512 pixels were randomly extracted and 160 images were used for training while 16 images were used for validation.

Figure 2 shows an overview of the in-house AI-integrated Dual BF ISH application. The application detects the nuclei in the selected ROIs and counts HER2 and CEP 17 signals. The automated quantification method calculates the total number of nuclei, HER2/CEP17 ratio, the number of HER2 and CEP17 signals per nucleus, their ratio to the nucleus, and ASCO/CAP ISH group class according to ASCO/CAP guidelines. The application allows users to remove or add nuclei and recount signals. The application was designed to select nuclei that contain at least two CEP17 and HER2 signals each to choose nuclei possessing complete two chromosomes 17. The method allows changing pre-defined thresholds for aneusomy, monosomy, and polysomy cases. Reported nuclei for assessment can be counted starting from 20 and up to 200 and verified by the reviewer. We analyzed 20 nuclei for regular cases or 40 nuclei for borderline cases.

### 2.6. HER2 Status and ASCO/CAP ISH Group Diagnosis

The HER2 test results are reported by the ASCO/CAP guideline recommendations for HER2 testing in breast cancer. The HER2 status is interpreted as amplified, based on a HER2/CEP17 ratio ≥ 2.0 and an average HER2 copy number ≥ 4.0 signals per cell [4]. We defined groups according to guidelines; HER2/CEP17 ratios ≥ 2.0 and average HER2 gene copies per cell ≥ 4.0 is Group 1, ratios ≥ 2.0 and average copies < 4.0 is Group 2, ratios < 2.0 and average copies ≥ 6.0 is Group 3, ratios < 2.0 and average copies ≥ 4.0 and <6.0 is Group 4, ratios < 2.0 and average copies < 4.0 is Group 5 [20,21].

### 2.7. Performance Analysis

This study compared the scanning protocols in terms of scan time, image quality, and impact on analysis results of the in-house AI-integrated Dual BF ISH application. The scan time depends on the tissue size and the image resolution. For a more accurate interpretation, we compared the scan time of three randomly chosen different tissues under five different scanning protocols of Scanners A and B. The scanning protocol C1 could not be incorporated into the comparison because the image format of Scanner C does not have information about the scan time on its viewer. For image quality, corresponding areas are extracted from the WSI digitized with each scanning protocol and compared qualitatively. For the analysis result by the application, automated and manual quantification times, number of detected nuclei, and concordances of HER2 status and ASCO/CAP ISH group with ground truth were compared quantitatively.

### 2.8. Ground Truth of HER2 Status and ASCO/CAP ISH Group

In this study, the clinically reported HER2 status and ASCO/CAP ISH group by the manual counting of Dual FISH was used as a ground truth for experiments. We evaluated the concordance of the manual Dual BF ISH results on WSI and glass slides using a microscope with the manual Dual FISH result. At least 20 cells were counted by two pathologists. Two areas are counted if another area includes >10% HER2 amplified cells and at least 20 cells or more are counted for equivocal cases as ASCO/CAP guidelines recommend [20]. For one case without FISH result due to an IHC score of 3+, only the manual Dual BF ISH results were compared and were classified into a Group 1 category as amplified case (Case 6).

### 2.9. Statistical Analysis

Statistical analyses were performed with R statistical software (EZR, version 1.68, Jichi Medical University, Saitama, Japan) (Kanda, 2013) [22]. The Kruskal-Wallis test was used for multiple comparisons; the Bonferroni method was applied, for a post-hoc test.

## 3. Results

### 3.1. Scanning Time

Table 3 shows the scanning time on three representative tissues with the scanning protocols A1, A2, and B1–B3. A1 and A2 protocols took the same scan time. Since the scanning protocol B3 applied the extended focus mode, the scan time increased over the number of Z-stack layers. By comparison of A1, A2, and B1, even with the same objective lens magnification, the scan time of Scanner A is shorter than that of Scanner B, since the image resolution of Scanner B is 1.5 times higher than that of Scanner A.

### 3.2. Image Quality

Figure 3 shows the images scanned with each protocol for case 9 with a HER2/CEP17 ratio of 2.49 and 11.19 HER2 signals/cell. The scanning protocol A2 produced bright and rather faint images. In particular, both HER2 and CEP17 signals are clearly digitized and easily recognized by human eyes as indicated. However, the nuclei are pale and the border with the background tissue is less distinct. As for nuclei, the scanning protocols A1 and B1–B3 provide bold color and sharp borders. The scanning protocol B1, being the highest resolution, has the potential to acquire the clearest image. However, we observed a lot of blurring and out-of-focus areas resulting in unstable image quality at least in part. By comparing the scanning protocols B2 and B3, the 3-layer extended focus mode produces better and more stable image quality than the single-layer mode in Scanner B because the in-focus image planes were to be selected from multiple layers automatically. The images by scanning protocol C1 are more blurred and paler than those of protocol A2. Some of the cases with unacceptable artifacts and out-of-focuses were re-scanned but were not successful.

### 3.3. Number of Detected Nuclei by Automated Image Analysis in Application

The representative nuclei detected by each protocol in the same ROIs are shown in Figure 4 for case 5 with a HER2/CEP17 ratio of 1.73 and 6.65 HER2 signals/cell. The number of detected nuclei by the application for each scanning protocol is depicted in Figure 5A. The minimum number of detected nuclei was 20 and 15 in the protocols of Scanners A and B, respectively, and 0 with Scanner C. The maximum number was 184 in the protocol B3. The best average number per case [average ± standard error (AVE ± SE)] was 80.1 ± 16.4 for the scanning protocol B3, showing the clearest nucleus borders and the most vivid signals, followed by 74.6 ± 16.0 for A1 and 65.7 ± 13.2 for B2. B1 possesses 40× objective and the highest resolution was 53.5 ± 11.8 including one case less than 20. A2 with a water immersion lens was 46.2 ± 9.8. The worst was 11.9 ± 3.1 for C1. The application could not well detect the nuclei on the WSIs with protocol C1, because of the pale nuclei, unclear nuclei borders, and weak signal appearance. Distinct nucleus borders and clear vivid signals are important for each detection method. Protocol C1 showed significantly lower nuclear detection numbers compared to the other protocols (Kruskal-Wallis test, *p* < 0.0002).

In addition to the initial selection of ROI, additional annotations were required until at least 20 nuclei were detected to meet the ASCO/CAP Guideline (Figure 5B). Protocols B1 and C1 needed more ROIs to get accurate results as compared to other protocols. The average file size selected as ROIs were, 2.82 ± 0.43, 2.21 ± 0.45, 5.74 ± 0.99, 1.69 ± 0.23, 2.50 ± 0.69, and 10.2 ± 1.20 mega bites (MB) (AVE ± SE) in the A1, A2, B1, B2, B3, and C1 protocols, respectively. As seen, the C1 protocol needed more ROIs to detect the nuclei although resulting in unsatisfactory consequences. In the protocols A1, A2, and B3, ROI sizes were reasonably low to obtain 100% accurate results. B1 with the highest resolution among all the protocols required more ROI because of unstable focusing.

The number of detected nuclei per unit MB of ROI (Figure 5C) would simulate the efficiency of the calculation of the program. The figures are 29.5 ± 6.2, 27.2 ± 6.2, 11.8 ± 2.7, 42.2 ± 9.1, 45.4 ± 10.1, 1.3 ± 0.4 (AVE ± SE) in the A1, A2, B1, B2, B3, and C1 protocols, respectively. C1 resulted in a tremendously lower number.

### 3.4. HER2 Status and ASCO/CAP ISH Group

Table 4 shows the analytical results by the application and the concordance with ground truth. The concordance of HER2 gene amplification status with the ground truth was 100% with the scanning protocols A1, A2, and B1–B3, and 20% with the protocol C1. ASCO/CAP ISH group concordance was 100% with the scanning protocols A1, A2, and B3, 90% with the protocols B1 and B2, and 10% for the protocol C1. The scanning protocol C1 resulted in the lowest concordance in both the HER2 status and ASCO/CAP ISH group because the nuclei segmentation or signal detections failed in 6 out of 10 cases. One ASCO/CAP ISH group from the Dual FISH disagreed with the manual Dual BF ISH on both WSI and microscope by two pathologists. This result confirmed that the Dual BF ISH WSIs that were collected for this study have the potential to provide the same results as the manual Dual FISH.

## 4. Discussion

We evaluated 6 different scanning protocols to assess our in-house application to evaluate HER2 gene status with the Dual BF ISH method utilizing 3 WSI scanners. The experimental results showed the importance of the image quality of the WSI including image contrast and the stability of image focus rather than just increasing the resolution. Automated nuclei detection can fail when image quality is suboptimal—such as in blurry cases or those with noise. In the present study, reduced efficiency in nuclei detection may be attributable to ambiguous or blurred nuclei in Protocols A2 and C1, as suggested by Hossain et al. [9].

We analyzed the borderline cases with a count of 20 or 40 nuclei and recorded the number of nuclei detected by the application in the selected ROIs. The result of the nuclei detection in the application depends on the image quality. Our study showed that the best results were observed in the scanning protocols A1 and B3 which provide the highest average number of detected nuclei 74.6 ± 16.0 and 80.1 ± 16.4, respectively. More than 20 nuclei were detected across most of the cases scanned by Scanners A and B, which is required in routine cases. Initial guidelines recommended evaluating at least 60 cells in mosaic and scattered patterns in tumors with HER2 heterogeneity [23]. In addition to the benefit of saving time using the in-house application for this laborious manual counting process, the scanning protocols A1 and B3 would make it easier to achieve the recommended cell counts for heterogeneous cases. Another study showed that counting 80 nuclei for a case for BF ISH HER2 assessment is more accurate for ISH Grouping [24]. Considering that the scanning protocol B3 could detect an average of 80.1 nuclei per case, it would be the best protocol for image analysis applications.

The regular single-layer mode often resulted in out-of-focus areas, and if it was serious, a re-scan was required to acquire an image with sufficient quality. For a stable scan, the Z-stack, which scans multiple layers along the depth direction, is a viable method, but it comes at the cost of increased scan time and file size depending on the number of layers. In addition, a larger file needs huge storage space, higher network bandwidth, and workstation memory capacity. In a previous study [25], the extended focus was explored as a promising alternative to the Z-stack. Since the extended focus also scans multiple layers, the scan time is almost the same as the Z-stack. The study reported that although the extended focus generally improved blurred areas, there was a loss of sharpness in the extended focus image compared to the multi-layer stacking image by the Z-stack. In this study, the extended focus mode was applied at the 20× magnification, and its scan time was longer than even the single-layer mode at the 40× magnification. On the other hand, the extended-focus image provided a higher concordance of the analytical result by the application and the shortest total quantification time. As the previous study mentioned increasing the layers for the extended focus may degrade the sharpness, the image quality of the provided extended-focus image was pretty good since this study employed only three layers, which is much fewer than the 10–16 layers the previous study utilized. Therefore, the scanning protocol B3 which applies the extended focus mode led the accurate nuclei detection and signal counting and less re-scanning and ROI re-selection.

Another benefit of the B3 and A1 protocols is that they have 100% accurate status results by AI-integrated applications without any human correction. Furrer et al.’s study validated a new classifier for automated analysis of FISH HER2 amplification in breast cancer and overall agreement between manual scoring yielded 98.4% before human correction [12]. Similar to the analysis software presented here, in one study the user had the possibility to manually correct the results obtained with automatic nucleus segmentation. They found a very good overall agreement (92.8%) between the results obtained with manual scoring by an expert and the results obtained with the image analysis software [26]. Automatic counting was performed in another study with an algorithm written in macro language using ImageJ (version 1.54p), an open-source image processing tool widely used for biomedical image processing. The algorithm works by dividing the image into red, green, and blue color channels and creates a nucleus-mask image to calculate the CEP17 and HER2 signals. The results showed that the corresponding agreement was 90% and the kappa value was 0.82 [13].

The study of Raimondo et al. studied 20 patient cases and the algorithm focused not only on detecting FISH spots per image but also on cell nuclei segmentation and case classification. Their results for Case-Based classification on the 12 testing FISH cases showed the ability of the proposed system to distinguish between all HER2 positive and negative assessments [27]. The study of Stevens et al. evaluated the automated signal enumeration system (Vysis Auto Vysion System) with the Vysis Path Vysion HER2 FISH assay; the concordance of the automated system was found to be 98% after excluding the 1.3–3.0 ratio [28].

All ROIs were selected in the same invasive areas for every WSI in the same tissue for each scan protocol. However, in some of the scan protocols, the same ROIs could not be used. The reason is not having the same ROI dimension and number was due to the lack of the same quality in the same region between the scan protocols, which prompted the pathologist to either make a smaller annotation in the same area or add another ROI in the same region. That could cause the difference between annotation numbers and sizes in the scanning groups.

One discordant case (Case 4) between Dual BF ISH in protocols B1 and B2 and manual Dual FISH was observed in the ASCO/CAP Group 4 case. In spite that there is a difference between the manual two methods with the human eyes, the A1 and B3 protocol results by the in-house application are 100% concordant with manual Dual FISH results. It has been shown that AI-integrated applications could provide a single and more accurate result in group determination when proper protocol is selected. Our study showed that digital pathology tools with new developing technologies are useful as mentioned in the literature [29,30,31].

Our study showed that it is essential to use an appropriate scanning protocol to obtain reliable results from the HER2 assessment application. The scanning protocol impacts the image quality, the number of detected nuclei, and the number of captured signals, which affects group determination and subsequent HER2 assessment results. The selection of the protocol also affects the duration of the scanning process and the turnaround time of HER2 judgment. We hope our study could be beneficial for other institutions that may have a plan to use AI-integrated ISH for HER2 assessment.

## 5. Conclusions

Selecting the scanning protocol is crucial while working with an image analysis tool. We believe our results could assist other institutions in the process of optimizing their scanning protocol.

## Figures and Tables

**Figure 1 bioengineering-12-00569-f001:**
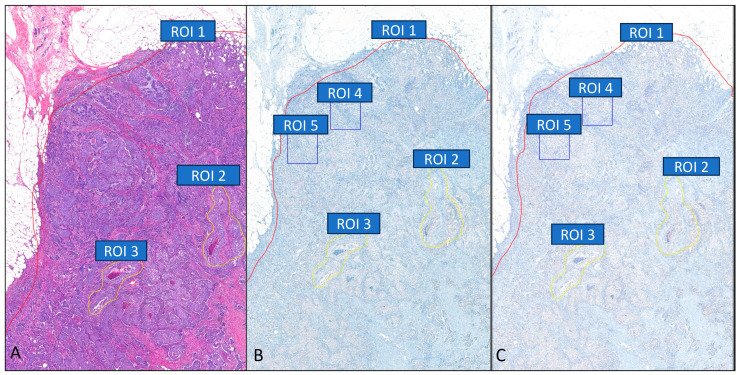
H&E and HER2-Dual BF ISH images by different scanning protocols at the same window at 4× high magnification (**A**–**C**). After an invasive area (red-ROI 1) and in situ areas (yellow-ROI 2 and 3) are drawn manually on H&E images (**A**), the annotations are cloned to the Dual BF ISH slide, and then a specific region of interest is selected (blue-ROI 4 and 5) for calculation on the Dual BF ISH slides (**B**), then cloned the other protocols (**C**).

**Figure 2 bioengineering-12-00569-f002:**
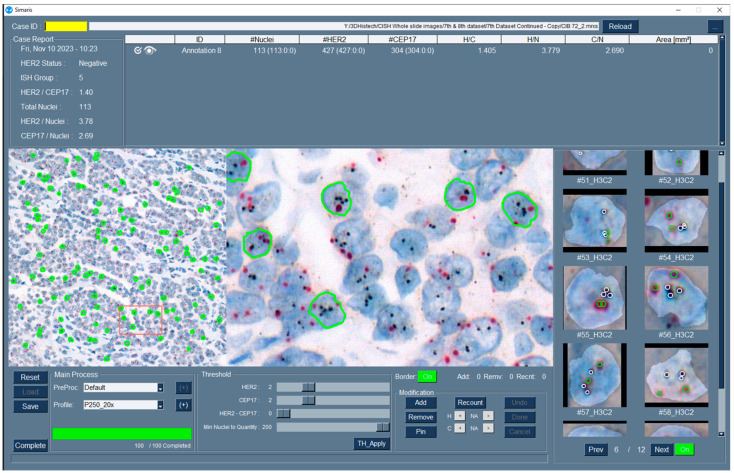
AI-Integrated Dual BF HER2 ISH quantification application. Application analyzes selected ROIs following the ASCO/CAP guideline recommendations for HER2 testing in breast cancer. Green solid lines on the left image represent detected nuclei. Up to two hundred nuclei can be reported using the threshold toolbar. The center image in the application shows zoomed-in detected nuclei. Nucleus selection can be modified by adding and removing and HER2 and CEP17 signal counts can be modified by recounting using the modification toolbar. Modifications for each nucleus can be accomplished by selecting images from the right side (HER2: white circles and CEP 17: green circles).

**Figure 3 bioengineering-12-00569-f003:**
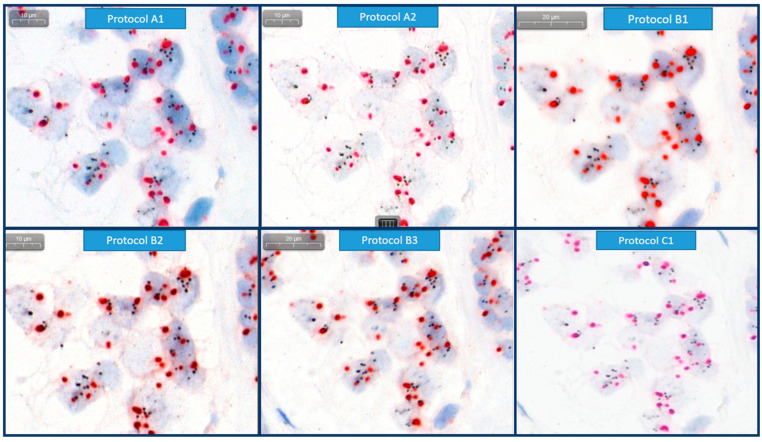
Example of image quality, nucleus borders, HER2 and CEP17 signals for each protocol in the same ROI of Case 9 (HER2/CEP17 ratio is 2.49 and HER2 signals/cell is 11.19 by manual FISH).

**Figure 4 bioengineering-12-00569-f004:**
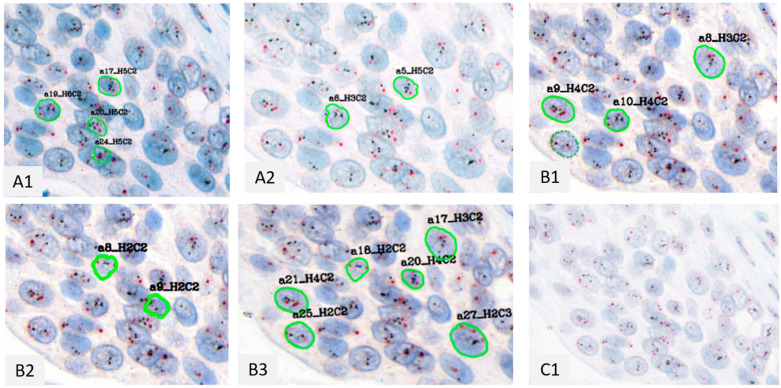
Nucleus detection by the application in the scanning protocols for the same ROI in Case 5. Each image is labelled by their protocol’s number ((**A1**): A1 Protocol, (**A2**): A2 Protocol, (**B1**): B1 Protocol, (**B2**): B2 Protocol, (**B3**): B3 Protocol, (**C1**): C1 Protocol). Green lines represent detected nuclei; The number of HER2 and CEP17 signals in each detected nucleus is calculated and the letters H and C represent these signals.

**Figure 5 bioengineering-12-00569-f005:**
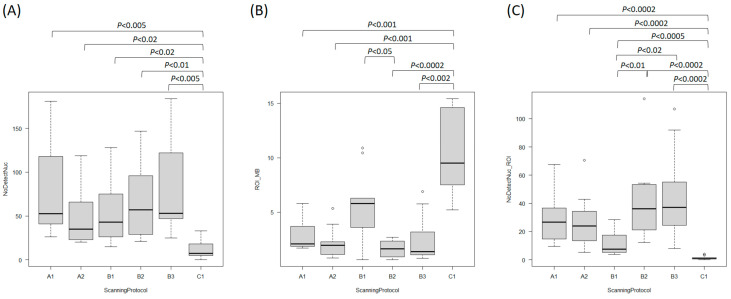
Box and whiskers plot showing the detected nuclei in selected ROI in each scanning protocol. (**A**) The total number of detected nuclei. Significantly different (Kruskal-Wallis test, *p* < 0.0002). (**B**) Necessary ROI (unit: mega bite, MB) to count as low as 20 nuclei. Significantly different (Kruskal-Wallis test, *p* < 0.00002). (**C**) The numbers of detected nuclei per one MB of ROI. Significantly different (Kruskal-Wallis test, *p* < 0.000002). X-axes show the scanning protocols. The Box and Whisker Plot displays the data distribution, with the box representing the interquartile range (Q1 to Q3) and the line inside indicating the median (Q2). Whiskers extend to data points within the 1.5× interquartile Range, while circle represent outliers beyond this range. Pairwise comparisons using Mann-Whitney U test are shown in the Figure.

**Table 1 bioengineering-12-00569-t001:** Dataset summary.

Case ID	Sample	Diagnosis	Primary vs. Met	AMP	IHC Score	FISH Ratio	HER2/Nucleus	CEP17/Nucleus	ISH Group
1	BX	IDC with lobular growth pattern	Primary	NA	1+ to 2+	1.41	3.82	2.7	5
2	BX	IDC involving dermis and Muscle	Met	NA	2+	1.31	2.36	1.8	5
3	EX	IDC with focal micropapillary features	Primary	NA	1+ to 2+	1.14	2.62	2.3	5
4	EX	ILC with focal pleomorphic features	Primary	NA	1+ to 2+	1.48	4.11	2.8	4
5	EX	IDC	Primary	NA	2+	1.73	4.64	2.7	4
6	EX	IDC	Primary	A	3+	n/a	n/a	n/a	1
7	BX	IDC	Primary	A	2+	3.89	7.18	1.8	1
8	EX	IDC	Primary	A	2+	2.65	7.2	2.7	1
9	EX	IDC with focal micropapillary features	Primary	A	1+ to 2+	2.49	11.19	4.5	1
10	BX	IDC, with focal micropapillary features	Primary	A	3+	8.88	18.65	2.1	1

BX: Biopsy, EX: Excision, IDC: Invasive ductal carcinoma, not otherwise specified, Met: Metastasis, AMP: HER2 Amplification, NA: HER2 amplification is not detected, A: HER2 amplified, IHC: HER2 Immunohistochemistry, n/a: not applicable.

**Table 2 bioengineering-12-00569-t002:** Scanning protocols.

	Scanning Protocol					
Features	A1	A2	B1	B2	B3	C1
Magnification of the objective lens	40×	40×	40×	20×	20×	40×
Numerical aperture	0.95	1.2 *	0.95	0.8	0.8	Unknown
Resolution (µm/pixel)	0.12	0.12	0.08	0.17	0.17	0.26
Z-stack	None	None	None	None	Extended focus of 3 layers at 1.4 µm interval	None

* Water immersion lens; Dry lens unless otherwise stated.

**Table 3 bioengineering-12-00569-t003:** Scanning Time (Seconds).

Case ID	Tissue Areas (mm^2^)	Scanning Protocol			
		A1&2	B1	B2	B3
1	140	65	106	34	229
7	280	145	175	42	703
4	473	204	272	54	863

**Table 4 bioengineering-12-00569-t004:** HER2 status and ASCO/CAP ISH group results by AI-integrated application and concordance with ground truth.

Case ID	Scanning Protocol												Manual FISH (Ground Truth)		Manual Dual BF ISH	
	A1		A2		B1		B2		B3		C1					
	G	S	G	S	G	S	G	S	G	S	G	S	G	AMP	G	AMP
1	5	N	5	N	5	N	5	N	5	N	5	N	5	NA	5	NA
2	5	N	5	N	5	N	5	N	5	N	n/a	n/a	5	NA	5	NA
3	5	N	5	N	5	N	5	N	5	N	n/a	n/a	5	NA	5	NA
4	4	N	4	N	5	N	5	N	4	N	5	N	4	NA	5	NA
5	4	N	4	N	4	N	4	N	4	N	n/a	n/a	4	NA	4	NA
6	1	P	1	P	1	P	1	P	1	P	n/a	n/a	1	A	1	A
7	1	P	1	P	1	P	1	P	1	P	n/a	n/a	1	A	1	A
8	1	P	1	P	1	P	1	P	1	P	5	N	1	A	1	A
9	1	P	1	P	1	P	1	P	1	P	5	N	1	A	1	A
10	1	P	1	P	1	P	1	P	1	P	n/a	n/a	1	A	1	A
Concordance with Manual FISH %	100	100	100	100	90	100	90	100	100	100	10	20				

G: Result of ASCO/CAP ISH group, S: Result of HER2 status, N: Negative, P: Positive, AMP: HER2 amplification result, A: HER2 amplified, NA: Non amplified, n/a: Not applicable.

## Data Availability

The datasets used during the current study are available from the corresponding author upon reasonable request.

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
