# Peer review of "Optimization of Scanning Protocol for AI-Integrated Assessment of HER2 Dual Bright-Field In-Situ Hybridization Application in Breast Cancer"

_bioengineering, 2025, doi:10.3390/bioengineering12060569_

Round 1
Reviewer 1 Report
Comments and Suggestions for Authors The article provides a detailed introduction to the importance of Human Epidermal Growth Factor Receptor 2 (HER2) in breast cancer treatment, as well as the advantages and disadvantages of current methods for detecting HER2 status, such as immunohistochemistry and in situ hybridization. It particularly points out that although Fluorescence In Situ Hybridization (FISH) is the gold standard for detecting HER2 amplification, its limitations, including the need for expensive equipment and the tendency of fluorescent signals to fade, have prompted researchers to seek alternative methods. Dual Bright-Field In Situ Hybridization (Dual BF ISH) serves as an alternative, offering the advantages of using conventional bright-field microscopy and providing permanent records. However, it requires pathologists to manually count signals, which is time-consuming. Therefore, combining Whole Slide Imaging (WSI) technology with Artificial Intelligence (AI) for image analysis applications holds promise for automating pathological diagnosis and improving work efficiency. Although the study selected 10 cases of breast cancer, the relatively small sample size may limit the universality and statistical power of the results. In future studies, it is recommended to increase the sample size to further verify the performance of different scanning protocols. The assessment of image quality relies to some extent on subjective judgment. Although the study conducted a qualitative analysis by comparing images from different protocols, it lacks more objective quantitative indicators. It is suggested to introduce more quantitative indicators, such as signal-to-noise ratio, contrast, and resolution, to more comprehensively evaluate the impact of scanning protocols on image quality. In summary, it is recommended that the authors make major revisions to the manuscript and improve the data.Author Response
Comments 1…Although the study selected 10 cases of breast cancer, the relatively small sample size may limit the universality and statistical power of the results. In future studies, it is recommended to increase the sample size to further verify the performance of different scanning protocols
Response-1. Thank you pointing this out. We agree that case number is less, but evaluation of this AI-based application is made by our previous work for Dual Bright field ISH accuracy, and this study focuses more protocol selection. Six different protocols were used for 10 cases and a total of 60 WSIs were created, from which 461 ROIs were selected, uploaded, analyzed and evaluated. Fifty of the 60 WSIs were scanned by scanner A and B protocols and 243 specific ROIs, averaging 3-4 ROIs per WSI, were selected by pathologists and analyzed by the AI application. The remaining 10 of the 60 WSIs were scanned by scanner C protocol, and 218 specific ROIs, averaging 10-20 ROIs per WSI, were selected and analyzed.
Number of WSI and ROI evaluation is added to methods section to be able to show the evaluated sample size is actually not that small. The change can be found in the revised manuscript- page number 3, 2nd paragraph and line 95-100.
Comments-2. The assessment of image quality relies to some extent on subjective judgment. Although the study conducted a qualitative analysis by comparing images from different protocols, it lacks more objective quantitative indicators. It is suggested to introduce more quantitative indicators, such as signal-to-noise ratio, contrast, and resolution, to more comprehensively evaluate the impact of scanning protocols on image quality.
Response-2. We appreciated for this comment. Automated nuclei detection can fail when image quality is suboptimal—such as in blurry cases or those with noise. In the present study, reduced efficiency in nuclei detection may be attributable to ambiguous or blurred nuclei in Protocols A2 and C1, as suggested by Hossain et al. (Hossain, 2022), although whole slide images (WSIs) were not assessed for image quality. It may be difficult to reanalyze image quality for all the WSIs in time. Our pupose is more to compare HER-2 ISH automated resuts with manuals.
This information is added in discussion section and can be found page number: 10, 1st parapraph, line 316-320
Reviewer 2 Report
Comments and Suggestions for Authors
This paper proposed an AI-integration to optimize the scanning protocol. Although the work is interesting and useful, there are some suggestions for further improvements:
1) On page 2, it is stated that "Several AI-integrated analysis applications have been applied for ISH quantification." Therefore, how the performance of the proposed AI-integration in this paper as compared by those works? The authors should show some comparative results by comparing their results with the methods by other researchers.
2) In Table 2, for "Numerical aperture", it is "0,95" or "0.95"? Please check also for "Resolution".
3) The main contribution of this paper should be "AI-integration". But the paper does not explain clearly on what type of AI has been used (e.g., fuzzy logic, SVM, neural networks, etc.).
4) In the caption of Figure 2, please avoid from using "you".
5) For the AI development, how many images have been used as the training, and how many have been used for validation and testing?
Author Response
Comment-1: On page 2, it is stated that "Several AI-integrated analysis applications have been applied for ISH quantification." Therefore, how the performance of the proposed AI-integration in this paper as compared by those works? The authors should show some comparative results by comparing their results with the methods by other researchers.
Response-1: Thank you for the recommendation. The comparison between three software is added in the discussion section, can be found in the revised manuscript- page number 11, 2nd and 3rd parapraph and line 358-371
Comment-2: In Table 2, for "Numerical aperture", it is "0,95" or "0.95"? Please check also for "Resolution".
Response-2: Thank you for catcthing. The numbers are edited in correct way. The change can be found in the revised manuscript- in the Table 2. page number 5
Comment-3: The main contribution of this paper should be "AI-integration". But the paper does not explain clearly on what type of AI has been used (e.g., fuzzy logic, SVM, neural networks, etc.).
Response-3: In our study, we used the deep learning AI-based application that was proposed as an in-house application in our department in the previous study and that enables the ex-traction and selection of non-handcrafted features by a convolutional neural network (CNN). This application applies a non-linear support vector machine (SVM) binary clas-sifier to select individual cores using nine shape and boundary-based features. For nu-clear detection, 10-fold cross-validation was performed using 3020 cores from 15 different cancer regions in 7 WSIs, each from a different cancer case . For this study CNN is used for the segmentation of the nucleus and each signal. The basic network architecture is U-Net which is one of the CNN-based architectures developed for image segmentation. Convolution layers and activation layers on the typical U-Net were modified by the en-gineer. Ten WSIs with annotation were used. From each of them 176 images with 512x512 pixels were randomly extracted and 160 images were used for training while 16 images were used for validation.
Information is added in the revised manuscript- page number5; 1stparapraph, and line147-158
Comment-4: In the caption of Figure 2, please avoid from using "you".
Response-4: Thank you for the comment. Edited version can be found in the revised manuscript- in the caption of Figure 2. page number 6, line 175-179
Comment-5: For the AI development, how many images have been used as the training, and how many have been used for validation and testing?
Response-5: In our study, we used the deep learning AI-based application that was proposed as an in-house application in our department in the previous study and that enables the ex-traction and selection of non-handcrafted features by a convolutional neural network (CNN). This application applies a non-linear support vector machine (SVM) binary clas-sifier to select individual cores using nine shape and boundary-based features. For nu-clear detection, 10-fold cross-validation was performed using 3020 cores from 15 different cancer regions in 7 WSIs, each from a different cancer case . For this study CNN is used for the segmentation of the nucleus and each signal. The basic network architecture is U-Net which is one of the CNN-based architectures developed for image segmentation. Convolution layers and activation layers on the typical U-Net were modified by the en-gineer. Ten WSIs with annotation were used. From each of them 176 images with 512x512 pixels were randomly extracted and 160 images were used for training while 16 images were used for validation.
Information is added in the revised manuscript- page number5; 1stparapraph, and line147-158
Reviewer 3 Report
Comments and Suggestions for Authors
The paper presents a benchmarking study of a scanning protocols and scanners. While the research is generally well-structured and methodologically sound, several minor revisions are needed before the manuscript can be considered for publication.
- Detailed technical specifications and descriptions of scanners A, B, and C should be included. This would serve as a valuable reference point for readers and improve the reproducibility of the study.
- The caption for Figure 3 is overly similar to the accompanying text in the manuscript. One of these should be revised to avoid redundancy and enhance clarity.
- The in-text citation format should be updated to conform to the journal’s specified referencing style.
- Incorporating a comparison with other similar scanning software that yields comparable results would strengthen the case for a standardized quality scanning protocol and provide additional context for the benchmark findings.
Author Response
Comment-1: Detailed technical specifications and descriptions of scanners A, B, and C should be included. This would serve as a valuable reference point for readers and improve the reproducibility of the study.
Response-1: Scanner A is an ultra-high throughput full-slide imaging scanner designed to meet the demands of large-scale laboratories and research institutions. It provides multi-layer scanning for better depth and clearer analysis of complex cytological samples. It provides ultra-sharp imaging with enhanced resolution and contrast, ideal for detailed applications such as cytology and immunohistochemistry. Scanner B is a high-speed digital slide scanner that offers a 300-slide capacity and supports both bright-field and fluorescence scanning. It features AI-powered texture detection and next-generation automation such as Z-stack scanning and autofocus scanning. The lens on Scanner C features extra-wide flat field correction, providing a much larger field of view (1 mm) that can accommodate extremely large digital images for fast scans and a resolution of 0.26 m/pixel at 40x magnification. Real-time focusing enables digital image capture that combines an im-aging line sensor and a focusing line sensor.
The change can be found in the revised manuscript- page number 4; 2nd parapraph, 130-141 line.
Comment-2: The caption for Figure 3 is overly similar to the accompanying text in the manuscript. One of these should be revised to avoid redundancy and enhance clarity.
Response-2: Thank you for catching redundancy. The caption and legend for Figure3 is updated. The change can be found in the revised manuscript- in Figure 3-caption and page number 7; 1st parapraph, 242-245 line.
Comment-3: The in-text citation format should be updated to conform to the journal’s specified referencing style.
Response-3 : It is done.
Comment-4: Incorporating a comparison with other similar scanning software that yields comparable results would strengthen the case for a standardized quality scanning protocol and provide additional context for the benchmark findings.
Response-4: Thank you for this comment. Comparison of the results is added in the discussion section, can be found in the revised manuscript- page number 11, 2nd and 3rd parapraph and line 358-371
Round 2
Reviewer 1 Report
Comments and Suggestions for Authors
I have no other problem.